# Urine-Derived Epithelial Cells as Models for Genetic Kidney Diseases

**DOI:** 10.3390/cells10061413

**Published:** 2021-06-06

**Authors:** Tjessa Bondue, Fanny O. Arcolino, Koenraad R. P. Veys, Oyindamola C. Adebayo, Elena Levtchenko, Lambertus P. van den Heuvel, Mohamed A. Elmonem

**Affiliations:** 1Department of Development and Regeneration, KU Leuven, 3000 Leuven, Belgium; tjessa.bondue@kuleuven.be (T.B.); fanny.oliveiraarcolino@kuleuven.be (F.O.A.); koenraad.veys@uzleuven.be (K.R.P.V.); christiana.adebayo@kuleuven.be (O.C.A.); elena.levtchenko@uzleuven.be (E.L.); bert.vandenheuvel@kuleuven.be (L.P.v.d.H.); 2Department of Pediatrics, Division of Pediatric Nephrology, University Hospitals Leuven, 3000 Leuven, Belgium; 3Centre for Molecular and Vascular Biology, Department of Cardiovascular Sciences, KU Leuven, 3000 Leuven, Belgium; 4Department of Pediatric Nephrology, Radboud University Medical Center, 6500 Nijmegen, The Netherlands; 5Department of Clinical and Chemical Pathology, Faculty of Medicine, Cairo University, Cairo 11628, Egypt

**Keywords:** podocytes, proximal tubular epithelial cells, PTECs, urine-derived cells, glomerular diseases, Fanconi syndrome, renal tubular acidosis, inherited renal disorders, kidney disease cellular models

## Abstract

Epithelial cells exfoliated in human urine can include cells anywhere from the urinary tract and kidneys; however, podocytes and proximal tubular epithelial cells (PTECs) are by far the most relevant cell types for the study of genetic kidney diseases. When maintained in vitro, they have been proven extremely valuable for discovering disease mechanisms and for the development of new therapies. Furthermore, cultured patient cells can individually represent their human sources and their specific variants for personalized medicine studies, which are recently gaining much interest. In this review, we summarize the methodology for establishing human podocyte and PTEC cell lines from urine and highlight their importance as kidney disease cell models. We explore the well-established and recent techniques of cell isolation, quantification, immortalization and characterization, and we describe their current and future applications.

## 1. Introduction

Expansion of urine-derived epithelial cells in vitro was developed almost 50 years ago [1,2,3]. Although the initial focus for such research was mainly to study cancer cells originating from the urinary tract [4,5], it expanded rapidly to cover all types of kidney disorders, particularly genetic kidney diseases, as cells isolated from urine carry the genotypic background of patients [6,7,8,9]. The easy and non-invasive approach of harvesting epithelial cells from urine has lured scientists to master cell expansion in culture to use them as a powerful tool to study the pathogenesis of various renal diseases and to screen for new therapeutic modalities [10]. Recently, the potential therapeutic effect of some of the isolated urinary cells has been even suggested [11].

Epithelial cells voided in human urine can include cells anywhere from the urinary tract and kidneys. In women, vaginal epithelial cells are also normally shed in urine. Fine adjustment of culture conditions is necessary to properly isolate and expand each type of the target epithelial cells. Next, in depth characterization of viable cells lost in urine helps to identify their origin by studying the expression of cell surface markers, gene and protein expressions and their functionality [12].

For the study of genetic kidney disorders two types of epithelial cells stand out: podocytes forming the glomerular tuft and proximal tubular epithelial cells (PTECs) layering the proximal part of the renal tubules. Both cell types are actively involved in the pathogenesis of numerous inherited kidney disorders [10]. This review presents the recent and the most well-established techniques for isolation, handling and characterization of these cells and some applications of these cellular models for the study of genetic kidney diseases.

Podocytes are the main cellular components of the renal glomerular filtration barrier. They are highly differentiated cells preserved throughout the animal kingdom with the main function of retaining plasma proteins and blood cells [13]. The glomerular filtration barrier is responsible for the selective filtration of blood coming through the afferent arteriole to the Bowman’s space. The glomerular capillaries are lined with a fenestrated endothelium sitting on the glomerular basement membrane (GBM). The glomerular endothelium is covered by the glycocalyx, a surface layer comprising glycosaminoglycans, proteoglycans, glycoproteins and glycolipids that contribute to the glomerular permselectivity [14]. Podocytes are attached to the GBM via highly organized foot processes [15]. In between the foot processes are the slit diaphragms (Figure 1), which are the final barrier preventing the passage of proteins into the urinary filtrate.

The filtrate that passes into the Bowman’s space continues into the proximal tubule and loop of Henle, distal tubules and collecting ducts for further processing [16].

There are currently over 60 genes, which produce important proteins that are reported to affect podocyte function and cause human glomerular disease when deficient. Pathogenic variants in these genes alter the dynamics of the glomerular filtration barrier changing podocyte morphology and their natural connection to each other, usually causing what is known as podocyte foot process effacement associated with widening of the slit diaphragmatic spaces [17]. This leads to proteinuria, renal tubular toxicity, gradual kidney function deterioration and eventually kidney failure [18]. When entering the process of foot process effacement, the foot process retraction and the replacement of the sealing of the filtration slits by occluding junctions results in increased attachment of the podocytes to the basement membrane. However, failure of the occluding junction formation leads to hyperfiltration and an expansile force on podocytes, leading to their detachment [19,20].

Proximal tubular epithelial cells (PTECs) are responsible for the reabsorption of the majority of water and solute load that passes through the glomerular filtrate. This reabsorption process takes place through specialized channels and transporters that allow the active transport of ions and different compounds in one direction only from the luminal side of the glomerular filtrate to the basolateral side back to the plasma (Figure 2). Receptor-mediated endocytosis is another essential function of PTECs aiming at the reabsorption of selected proteins and other nutrients, including carrier-bound vitamins, amino acids and trace elements, which are filtered by the glomeruli [21]. The multi-ligand receptors megalin and cubilin, also present in other luminal organs, coordinate the uptake of most filtered proteins and other small bioactive molecules from the lumen (Figure 2). Within the kidney, megalin is expressed extensively at the apical brush border and in the apical endocytic compartments of epithelial cells that comprise the early segment of the proximal tubule, with decreasing expression as it goes distally. Cubilin is also expressed abundantly in the proximal tubule and usually interacts with megalin to increase the multiligand binding potential of the complex [22].

The sophisticated function of the renal proximal tubule in regulating electrolyte balance, acid–base balance and many essential proteins, made it prone to numerous genetic abnormalities [23]. Over 25 genetic diseases have been linked with proximal tubular pathology. Many of these disorders are collectively referred to as proximal tubulopathies or renal Fanconi syndromes (when a generalized proximal tubular dysfunction is present). The basic pathology is the inability of the proximal tubule to reabsorb critical molecules, such as bicarbonate, calcium, phosphate and several proteins and vitamins. These disorders are commonly associated with severe morbidities, as the dysfunction of the proximal tubule usually results in significant metabolic disturbances [24].

## 2. Quantification of Podocytes and PTECs in Urine

Quantification of urine-derived epithelial cells has emerged as a valuable non-invasive clinical assessment method for detecting, studying and monitoring kidney disease [25]. Quantifying cell number in kidney specimens and in urine has become an intensive area of research. Since urine can be obtained non-invasively and repeatedly during the disease course, it is especially attractive as a cell source.

There are on average 1 million nephrons in each of the two kidneys, but the actual measured number might vary from 210,000 to 2.7 million [26]. Increased numbers of podocytes have been found in the urine of patients suffering from different glomerular diseases, in comparison with healthy individuals. In model systems, it has been demonstrated that a complete glomerular collapse occurs when podocyte cell number falls below 20% of original number. Since a normal human glomerulus has on average 500 podocytes [27], loss of more than 400 podocytes per glomerulus can cause kidney failure [28]. Podocyte loss (podocyturia) has been described in various kidney pathologies such as diabetic nephropathy [29,30], membranous nephropathy, glomerulosclerosis, Henoch-Schönlein nephritis, IgA and Lupus nephritis, diffuse mesangial sclerosis, Alport syndrome, APOL1-related nephropathy and cystinosis [10,29,31,32,33,34,35], for which cell models have been developed.

The general principle of quantifying urine-derived kidney epithelial cells is based on the identification of genes and proteins specific to these cells in fresh urine samples. These specific genes and proteins are mainly podocalyxin, nephrin, podocin and synaptopodin, which are markers of podocytes and aquaporin-1, megalin, cubilin and CD13, the markers of PTECs [6,25,36,37,38,39].

There are several methods used for the quantification of urine-derived epithelial cells. These methods include: colorimetric quantitative methods [6,37], flow cytometry [36], Western Blot [25], mass spectrometry [40] and quantitative real-time polymerase chain reaction (qRT-PCR) [25,38].

### 2.1. Colorimetric Quantitative Methods

Through immunohistochemistry, immunofluorescence and immunoassays such as the enzyme-linked immunosorbent assay (ELISA), different proteins specific to the kidney epithelial cells are targeted with monoclonal antibodies to allow the quantification of epithelial cells via an appropriate method [41,42]. These methods are widely used in detecting urinary epithelial cells [43,44,45,46] and are considered the “gold standard” because of the highly specific interactions between the protein of interest and the targeted antibody, and also their ability to detect a very low amount of proteins. Although these colorimetric methods are widely used, they still come with some limitations, including unspecific binding as urine can contain several proteins, exosomes or extracellular vesicles and cell debris, thereby leading to examiner errors/bias [37]. Another major drawback of this technique is the production of antibodies which are usually cost and time consuming and the lack of commercially available antibodies for all isoforms of the protein, as in the case of podocin. Recent evidence suggested that podocin may exist in both a canonical, well studied large isoform and an ill-defined short isoform [47].

### 2.2. Flow Cytometry

In this technique, the epithelial cells are sorted by utilizing highly specific antibodies labelled with fluorescent conjugates [25,36]. Flow cytometry has been demonstrated to be a very reliable, sensitive and accountable method in assessing kidney pathologies [48]. In many glomerular diseases such as IgA nephropathy, diabetic nephropathy and focal segmental glomerulosclerosis, flow cytometry has been used with immunofluorescence stain with podocalyxin antibody [49,50]. Nevertheless, the use of this method in quantifying cells has been limited due to unspecific binding, cost of the machine and its consumables and the high level of expertise required to operate the flow-cytometer. Many times, the low number of cells present in the urine sample makes sorting impractical.

### 2.3. Western Blot

This method is based on the use of specific monoclonal antibodies to detect a specific protein expressed in the target epithelial cells [25]. This method has been widely used in quantifying and detecting protein specific to kidney epithelial cells [6,8] because itis highly sensitive and specific even at a very low concentration (picogram level). However, the use of Western Blot is still limited partly due to the unspecific binding as a result of off-target interaction with other proteins, presence of exosomes and cell fragments, the lack of specific antibodies for all target proteins, especially the post-translational modified protein targets and high technical demand on the scientist [51].

### 2.4. Mass Spectrometry

The main principle behind this method is the use of tryptic peptides as surrogate markers for the quantification of their respective proteins. Therefore, by using high-performance liquid chromatography (HPLC) coupled with tandem mass spectrometry, one can easily detect a protein tryptic peptide specific for one of the kidney epithelial cells [40]. Mass spectrometry for quantifying urinary kidney epithelial cells was first demonstrated as a tool for detecting podocyturia in urine sediments of individuals with pre-eclampsia in 2012 [40]. In this study, Garovic et al. detected the podocin tryptic peptide in the urine of women with pre-eclampsia by using HPLC coupled with tandem mass spectrometry with the addition of an isotopically labelled podocin peptide [40]. Furthermore, efforts are currently being made towards using this technique to detect biomarkers of kidney epithelial cells in soluble fractions rather than in the urine sediment. Indeed, a research group in France has being able to quantify podocin and aquaporin-2 in human urine by using a new mass spectrometry approach, termed “liquid chromatography-multiple reaction monitoring cubed mass spectrometry [46,52].

The main advantages of this method are: it is operator-independent, highly reproducible method and considered highly sensitive. Therefore, it may be used for an earlier diagnosis of kidney pathologies compared to the other methods. Furthermore, this technique does not require the generation of antibodies (a problem associated with the colorimetry and Western Blot methods), can be easily multiplexed for simultaneous quantification of multiple proteins, and it is potentially applicable across species (provided the quantified peptides are conserved) [40]. Although this method has a great future potential, its main disadvantages include the high cost of the instrument, it is time-consuming and a high level of expertise is needed.

### 2.5. Quantitative Real-Time Polymerase Chain Reaction

Quantification of specific kidney epithelial cells markers mRNA in urine pellets by qRT-PCR has used to study various kidney diseases [6,8,53,54,55]. This methodology relies on the specific transcript expression of each cell type, and it is an indirect estimation of the number of cells present in urine normalized to calibration curves. The calibration curves are created by using known numbers of an established kidney epithelial cell line [6]. Some of the advantages of this technique are that it is potentially quantifiable, specific and sensitive and it can be multiplexed to quantify several mRNAs simultaneously [46].

Often, the number of cells present in urine is too low for good quality mRNA extraction using traditional techniques. Therefore, our group has recently optimized the methodology based on single-cell mRNA extraction protocols. Briefly, a calibration curve using known numbers of control human kidney-derived epithelial and mesenchymal cells was developed. To establish the calibration curve, cells were sorted by Fluorescence-Activated Cell Sorting (FACS) in a 96-well plate containing 4 mL of lysis buffer (0.2% TritonX-100 + RNase inhibitor) yielding a range of cells per well from 5 to 500 cells.

Next, we performed the Smart-seq2 protocol [56] allowing cDNA synthesis from a very low numbers of cells. We followed the protocol of Picelli et al. [56] up to PCR purification (step 26), and we ran18 PCR cycles in step 14 for pre-amplification of genes. Last, qPCR was performed using various housekeeping genes, which estimated the total number of cells. Using specific primers, we estimated the number of each kidney cell type shed in urine. The cycle threshold (ct) value for the specific targets achieved in the qPCR analysis was plotted in the calibration curve for extrapolation of the number of cells shed in urine, which was normalized to urine volume and urine creatinine values.

When quantifying cells from urine samples of patients with kidney pathologies, often more than 500 cells are present, therefore the cell pellets were re-suspended in phosphate-buffered saline (PBS) and diluted 100×. Although this method shows high specificity, overcomes bias and allows evaluation of samples containing very few cells, it still comes with its own limitations. It is an estimate and not an absolute quantification of cells, andis time consuming and requires specific expertise.

## 3. Isolation and Immortalization Techniques

Kidney epithelial cells have restricted proliferation capacity, therefore for establishing a cell line, they need to be immortalized prior to expansion.

### 3.1. Isolation of Kidney Epithelial Cells

Kidney epithelial cells can be obtained from two main sources. First, the isolation of cells from the kidney tissue itself, resulting in cell lines from different kidney structures. However, the number of cells isolated from a tissue biopsy may be limited, and the isolation procedure is invasive and laborious. An alternative cell source is urine, which is the focus of this review. Both healthy individuals and patients suffering from kidney disease lose epithelial cells in urine [10]. Figure 3 shows a simplified scheme for the isolation of kidney cells from urine, the immortalization and expansion steps that follow. Isolating cells from urine is a rather simple and cost-effective process. Fresh urine is centrifuged at 200–300× *g* for 5 to 10 min at room temperature or 4 °C. The cell pellet is then re-suspended in the specific culture medium to selectively expand the targeted cells.

### 3.2. Immortalization of Isolated Cells

As mentioned, primary cells derived from kidney tissue or urine are terminally differentiated and show a very limited proliferation rate in vitro. Therefore, the immortalization step is fundamental for further expansion of a cell line.

In 2005, Wilmer et al. immortalized PTECs from the urine of cystinosis patients and healthy controls, using HPV E6/E7 [57]. However, the immortalizing activity of HPV E6/E7 is generally very weak in human epithelial cells. Furthermore, the cystinotic cell models that were generated with HPV immortalization did not accumulate sufficient cystine to accurately represent the desired phenotype due to their high proliferation rate [58,59].

Ex vivo infection or transfection of primary cell lines with the Simian virus 40 (SV40) has also been shown to increase the lifespan of both epithelial and non-epithelial cells. The small icosahedral virion with a double-stranded DNA genome is a member of the Polyomaviridae family and SV40-based immortalization methods have been applied extensively [60]. In 1965, a growth stimulatory effect of the SV40 DNA sequence on a BHK21 (Baby Hamster Kidney 21) cell line was reported. Wiblin and MacPherson later also successfully transformed this cell line by co-culturing with a monkey cell line that was infected with the SV40 virus. A detailed description of the immortalizing capacity of the SV40 came in 1984 by Chang et al. [61]. It was proven that a specific part of the SV40 DNA was responsible for the immortalizing capacity of the virus. More specifically, two proteins in the early region of the viral sequence can mediate cell transformation, the large tumor (T) antigen and small T antigen. While the small T antigen promotes efficient viral genome replication by accelerating both the G1 and S phase progression during cell division, the large T antigen can interact with growth suppressors. Notably, the large T antigen carries an LXCXE motif that induces interaction with three proteins from the tumor suppressor retinoblastoma (pRB) family [62]. More specifically, the large T-antigen binds to the pRb-E2F complex, resulting in the dissociation of E2 Factor (E2F) from the complex and activation of specific gene expression patterns inducing cell growth. Additionally, the large T-antigen can also suppress the p53-pathway, an important tumor suppressor protein, and thereby induce cell proliferation [63]. Subsequently, the blocking of the pRB- and p53-dependant tumor suppression activity will drive the quiescent cells to enter the S-phase and escape apoptosis [62].

High and uncontrolled large T antigen expression by SV40-transformed cells comes with some disadvantages. The main concern is that the continuous expression of an immortalizing gene can significantly alter normal cellular physiology. Because of this problem, a temperature-sensitive mutant form of the SV40 DNA, the SV40tsA58, was developed [64,65]. In this mutant form, the 708 amino-acid large T antigen segment of the SV40 carries a temperature sensitive mutation. More specifically, an Ala438Val mutation lies within the ATP-binding fold and results in the stable expression of the tsA58 large T antigen (LT) variant at the permissive temperature (32–33.5 °C), inducing cell growth. However, shifting the cells to the restrictive temperature (37–39 °C) reduces the stability of this tsA58 mutant T antigen and eliminates its ability to bind to p53, resulting in growth arrest in either the G1 or G2 cell cycle phase. Because of this characteristic, these cells are called conditionally immortalized cell lines [66,67,68].

The urine-derived conditionally immortalized cystinosis PTEC cell model that was developed in 1995 by Racusen et al. was immortalized by using a recombinant retrovirus with a pZipneoU19 plasmid that carried the temperature-sensitive mutant form of the SV40 T antigen allele [66]. In 2002, Saleem et al. also created a conditional immortalized podocyte model from kidney tissue by using the SV40tsA58. Therefore, these cells grow at 33 °C, but enter growth arrest and differentiate at 37 °C [69]. Later, in 2010, urine-derived cell lines from glomerulosclerosis patients were also established using the temperature-sensitive mutant of the SV40 large T antigen [70]. Furthermore, Orosz et al. generated three tissue derived proximal tubular cell lines by using either the wild-type or the temperature sensitive SV40 large T antigen [68].

Notably, the proliferation of human cells can be limited by telomere-dependent replicative senescence due to the end-replication problem, which entails that the ends of the lagging DNA strand cannot be completely replicated during the cellular division, resulting in the continued shortening of the telomeres during each subsequent mitotic cycle [71]. In 1998, the transduction of telomerase-negative human cells with the catalytic subunit of the human telomerase reverse transcriptase (hTERT) gene was shown to successfully extend the life-span of these cells by stabilizing the telomeres [72]. In 2008, Wieser et al. were able to immortalize tissue-derived PTECs by inducing an overexpression of hTERT. This cell line still showed the original differentiation status and functionality. The early passage hTERT-immortalized cells were shown to faithfully represent the physiological properties of the in vivo situation [73].

While hTERT can be used by itself, it is often combined with a temperature-sensitive SV40 resulting in conditionally immortalized cells that are also protected against the end-replication problem and the resulting senescence. Early on, co-expression of hTERT with p53 and pRB inactivation by either siRNA or SV40 large T antigen was shown to successfully immortalize primary cells [72]. Furthermore, a combination of hTERT with the SV40 large T antigen was used to successfully immortalize kidney cells [74,75]. Currently, this combination is the most prevalent and has been extensively applied to immortalize both tissue- and urine-derived podocytes and PTECs by several groups [6,10,70,76].

## 4. Molecular and Functional Characterization of Kidney Derived Cell Lines

Once cells have been immortalized, a homogenous cell line can be obtained by subcloning [77].Subcloning is usually done by using irradiated NIH 3T3 mouse fibroblast cells as non-dividing feeder cells. Briefly, cells are seeded at densities of 100, 200, 300 and 400 cells per 25 cm^2^ flask and grown at 33 °C. Subsequently, feeder cells are added to each flask at a density of 0.5 × 10^6^ cells/flask. After being cultured for about 21–28 days, clones derived from single cells become visible and are picked by using cloning discs drained in trypsin/EDTA and transferred to individual wells of a 24 well plate for expansion. At this stage, usually the fast-growing colonies are selected to be expanded further. Finally, when thecells are grown to confluence, they are further transferred to larger flasks, and each clone can be either cryopreserved or processed for characterization [69,76]. It is worth mentioning that subcloning of urinary kidney epithelial cells can also be done without feeder cells. Briefly, cells are seeded into a 96 well plate at densities of 0.5 and 1 cell per well. After 14 days, colonies derived from single cells become visible, and can be expanded [78].

Afterwards, clonal cell lines need to be characterized to identify the most potent and specific ones. First, podocytes and PTECs can be identified by their distinct morphology. Second, at the molecular level, podocyte- and PTEC-specific markers can be used to distinguish these two cell types based on their specific gene expression patterns. This can be done at two levels, at the mRNA and at the protein level. Marker analysis can be done at the mRNA level by analyzing the cell specific genes with specific primers or by RNA-seq analysis (transcriptomics). At the protein level, the analysis of the gene expression pattern can be done by Western blotting, immunohistochemistry and flow cytometric-based methods. Finally, a functional characterization study is based on the distinct function of each cell type and confirms the differentiation of the immortalized cell lines to functional PTECs or podocytes [79,80]. Figure 4 represents a simplified scheme for the main steps of the molecular and functional characterization of PTECs and podocytes.

### 4.1. Characterization of Podocytes

Podocytes have a very distinct morphology in vitro, showing a cobblestone-like or arborized morphology by bright-field and phase-contrast microscopy.

Podocalyxin (PDXL) was one of the first marker genes used to characterize podocytes, but is not sufficient to prove the cell identity alone, since this protein is also present on other epithelial cells. The same condition is valid for the use of tight junction proteins (ZO-1) or intermediate filament proteins (Vimentin) [80]. Therefore, a panel of expression is normally used, in which expression of several of these podocyte genes is identified.

One of the most specific proteins that can be used as a podocyte marker is nephrin (NPHS1), a key structural transmembrane protein of 1241 amino acids in the slit diaphragm. Furthermore, nephrin also recruits other slit diaphragmatic proteins such as podocin (NPHS2) and CD2 associated protein (CD2AP), both of which can also be used for characterization [81,82]. However, it has been shown that urinary podocytes can undergo irreversible dedifferentiation in artificial culture conditions, often leading to the loss of nephrin expression in these 2D cell lines [83]. Synaptopodin (SYNPO) is an actin-associated protein, important in the dynamic podocyte cytoskeleton [84]. Synaptopodin is a key marker for differentiated podocytes and can be detected both at the RNA- and protein-level to characterize podocytes [80]. Podocin (NPHS2) was first described in in vitro podocyte cell lines in 2002 and is currently one of the standard markers for podocytes [85]. Wilms’ tumor gene 1 (WT1) is a master regulator of podocyte gene expression, with reduced expression related to glomerulonephritis and mesangial sclerosis [86]. While its exact function is not yet completely understood, WT1 expression on the RNA- and/or protein level has already been used extensively for podocyte characterization [87]. Finally, filament type cytoskeletal proteins like nestin (NES) and β-tubulin (TUBB), while not podocyte specific, are also part of the slit diaphragm and can be included in characterization studies [70]. The glomerular endothelial cells, the glomeular basement membrane, and the filtration slits between the podocytes perform the filtration function of the glomerulus, separating the blood in the capillaries from the filtrate that forms in Bowman’s capsule (Figure 1).

The attachment of podocytes to the glomerular basement membrane is granted through various proteins and the production and assembly of collagen type IV α-chains (COL4A3, COL4A4 and COL4A5). The expression of collagen IV α3, α4 and α5 is podocyte specific and has been applied for the characterization of mature and functional podocyte cell lines [8,88].Additionally, in kidney organoids and 3D podocyte cultures, type IV collagen α-chains show higher abundance and are indicative of basement membrane formation [89,90,91,92,93].

Interestingly, the filtration function of the basement membrane can be modeled in vitro by glomerular permeability assays. In principle, this assay measures the amount of fluorescent-labelled albumin passing through a monolayer of podocytes and endothelial cells. Glomerular endothelial cells are usually seeded on the bottom side of the PET porous membrane embedded in the central part of a small bioreactor, while the differentiated podocytes are seeded on the upper side. After 2 days of setting the co-cultures, the perfusion assay is performed. Firstly, during the filtration assay, 1 mg/mL albumin–fluorescein isothiocyanate (FITC) conjugate medium or normal podocyte medium (1 mL) is added to lower glomerular endothelial cells compartment and the podocyte compartment of the bioreactor, respectively. After 3 h of perfusion, the transit of albumin–FITC conjugate from the lower glomerular endothelial cells compartment to the upper podocyte compartment in the experimental conditions is measured by collecting separately the liquid coming out of the two compartments of the bioreactor. Lastly, the FITC signal is then measured in triplicate by using a fluorimeter. This method has been used to validate podocytes developed from patients with Alport syndrome [8].

Another assay that can be utilized to assess the functionality of podocyte cell lines is the endocytosis of labeled albumin [94]. Furthermore, the specific cytoskeleton arrangement of podocytes can be analyzed by exposing the cells to angiotensin II in vitro, which leads to cytoskeletal rearrangements. Angiotensin II exposure will also result in a change in gene expression. Therefore the re-analysis of the above-mentioned markers after exposure to angiotensin II can also be used to assess the functionality of an in vitro podocyte cell line [6,94,95]. Calcium signaling is of key importance in the podocyte for several functions, including the maintenance of the structural integrity of the slit diaphragm. In general, calcium influx can be evaluated via a Fura assay and the functionality of the receptor potential cation channel, subfamily C member 6 (TRPC6), a marker for differentiated podocytes critical for calcium homeostasis, can be evaluated via a whole-cell patch clamp assay. Via this technology, the increase in calcium influx through TRPC6 in functional podocytes can be assessed using oleoyl-2-acetyl-sn-glycerol (OAG)a known agonist of TRPC6 [95]. Using one or a combination of these methods, various podocyte cell lines have been developed to model and control podocyte-associated kidney disease.

### 4.2. Characterization of PTECs

In vivo, PTECs are tall and cuboidal in shape with an apical brush border and basolateral invaginations. However, this specific morphology is not present in unpolarized cells in culture [79] and their morphological identification is observed by their spindle-like orientation.

Marker analysis of PTECs relies on the specifically expressed Aquaporin 1 (AQP1), often combined with the detection of cytokeratin (CK) 18 and γ-glutamyl transferase (GGT). However, the latter two are not specific for the kidney, but only indicate the epithelial origin. The mRNA or protein expression of CD13, the Na/K-ATPase and protein aminopeptidase N (ANPEP) have also been used to establish the epithelial origin of cells, as well as the expression of dipeptidyl peptidase IV (DPP4) and the multidrug resistance protein 4 (MDR4) [76]. Another method that can be used to establish the cellular epithelial origin is by combining an antibody staining against cytokeratin 7 (CK7), a marker for epithelial cells, and vimentin (VIM), which is a marker for mesenchymal cells. In this context, PTECs will be positive for cytokeratin 7 and negative for vimentin [96].

Lima et al. suggested that the expression of zonula occludens 1 (ZO-1)–associated nucleic acid binding protein (ZONAB) promotes proliferation, but represses the differentiation of these cell lines [97]. In order to confirm the polarization of PTEC cells for further characterization, two specific markers can be utilized. Immunostaining of the zonulin-1 (ZO-1) tight junction protein will reveal a grid-like structure in a confluent layer of PTECs, indicating a clear polarization and monolayer, while the microtubule end-binding protein (EB1) is also a marker of polarization [98,99,100,101].

When using a PTEC model, specific PTEC-functions need to be evaluated. The functionality of PTECs can be established by analyzing the epithelial transport, the main function of the proximal tubule in the excretion pathway. The uptake of albumin and phosphate can be used to assess the functionality of the endocytosis and sodium dependent uptake. Intracellular levels of albumin after endocytosis-mediated uptake in functional PTECs can be measured by flow cytometry or the uptake of a labelled albumin-fluorescein isothiocyanate conjugate can be analyzed by fluorescent imaging. Liquid scintillation counting can be used to assess the sodium dependent uptake of a labelled phosphate molecule (^32^PO_4_). Additionally, the functionality of the OCT influx proteins can be investigated by exposing the cells to the substrate: (4-(dimethylamino)styryl)-N-methylpyridiniumiodide (ASPþ) in the presence or absence of 5 mM OCT inhibitor tetrapentylammonium chloride (TPA). For P-glycopotein activity, the PTECs can be examined by measuring the accumulation of a fluorescence molecule that is formed by the intracellular cleavage of calcein-AM. Cells with a functional P-gp will rapidly extrude the calcein and reduce the accumulation, which can be assessed by fluorescence measurement. Transport characteristics of the ABCG2 and MRP4 renal efflux transporter can be assessed by liquid scintillation counting or imaging after incubating the cells with labeled kynurenic acid or with Chloromethylfluorescein-diacetate (CMFDA) respectively and an MRP inhibitor. GST-RAP or the transferrin endocytosis assay can be utilized to study the functionality of the megalin–cubilin receptor complex by analyzing the endocytosis process itself [102,103,104,105,106,107].

Functionality of the multiligand receptors that are responsible for endocytosis, megalin (LRP2) and cubilin (CUBN) are pivotal in PTEC lines. However, while cubilin often shows a clear expression at both the mRNA and protein level, the expression and functionality of megalin is not always as clear, due to the fact that an in vitro cell line will not always be polarized [102]. A similar problem has been seen with AQP1, while the staining of this channel in kidney tissue shows a clear apical distribution, in vitro cell lines usually show a more evenly distributed pattern due to the absence of polarization, underlining the important difference between 2D cell culture, and the in vivo situation [108].

When using PTEC models for drug testing, the expression of organic anion transporters (OATs) is often desired, including OAT1/SLC22A6 and OAT3/SLC22A8. However, OAT expression is often lacking in 2D cultures of immortalized PTECs, as is the case with HK-2 cell lines. Therefore, some OAT-overexpressing lines have been developed for the investigation of drugs, but the expression can also be ensured by using 3D cell cultures or primary human tubular cell monolayers, both of which have established OAT-expression [109,110,111,112]. Using one or a combination of these methodologies, various PTEC cell lines have been developed to model and control tubule-associated kidney disease.

### 4.3. The Choice of Control Cell Lines

Often, the viability and functionality of cell lines isolated from urine is questioned, but their effectiveness has been proven several times. Clear comparison of PTECs derived from tissue versus from urine showed similar functionality, as both expressed functional transporters and receptors, as well as similar physiology. This underlines that both isolation methods are equally successful in establishing cell models, while urine outstands due to non-invasive isolation and less laborious practices.

Patients suffering from kidney diseases are characterized by the highest degree of cell loss in the urine. However, healthy individuals also shed viable kidney cells in urine, and expansion of those applying the same method of isolation and immortalization as described above is possible and may serve as control to cells from renal patients [66]. Still, the generation of control cell lines is limited by the small number of viable cells [76]. It is important to consider that healthy donors are those that have not been diagnosed with kidney disease, which does not exclude the possibility of unknown damage leading to cells detachment, thus the processes of cell characterization and selection are extremely important to ensure the functionality of the urine-derived control cells before being used in experiments. In this regard, several studies have compared properly selected control cells from urine with cells isolated directly from the kidney and proved their similarities and functionality, underlining the potential of these cell lines as wild type controls for the study of models of kidney diseases [10,77].

It has been demonstrated that about 1.7 podocytes are lost per glomerulus per year in healthy humans [113], while this number is highly exceeded in disease conditions and during aging [114]. Wilmer et al. collected mid-stream urine of 38 healthy volunteers and found that 10% of the collected urine sediments contained viable cells with proliferative capacity in vitro. Based on the results from the characterization study on these proximal tubular cells, one clone from one donor was selected for further use. This study introduced the first human cell line that featured a functional sodium-dependent reabsorption and endocytosis for up to 40 passages [10,76].

Vogelmann et al. used the cytospin method to quantify the podocyte loss in patients with active glomerular disease and healthy volunteers. It was found that all but one of the healthy volunteers had a urinary podocyte content of <0.5 podocytes/mg creatinine. Furthermore, 12 of the 27 healthy individuals that were studied had podocytes in at least one of their urine samples. All of these volunteers had viable podocytes in their urine sediments, with 60% of the healthy controls growing podocytes in culture and the growth pattern resembling that of cell lines established from glomerular explants [115].

As an alternative for the limitations imposed by low proliferation of fully differentiated epithelial cells, our laboratory has developed kidney stem/progenitor cells lines derived from the urine of healthy preterm neonates [99] as a prototype of podocyte and tubular disease model. These cells have high proliferative capacity and can differentiate into fully mature and functional podocytes and PTECs in vitro, overcoming the immortalization process, maintaining the intrinsic properties of the donor and serving as control kidney epithelial cells.

### 4.4. 2D versus 3D Models

In vivo, cell–cell and cell–matrix interactions are important in controlling the cell phenotype and cell function. However, Two-dimensional (2D) cell cultures rarely reproduce these conditions, and the absence of the cellular microenvironment can hamper the in vitro cell model-based research. Three-dimensional (3D) cell culture systems and organoids have been manufactured for the kidney and have been shown to reproduce the kidney-specific functions more accurately than 2D models [116]. Furthermore, the generation of these 3D models can improve the ability to cure and manage kidney diseases, model the kidney development and could revolutionize the world of kidney replacement therapy by allowing the in vitro production of transplantable kidneys [117]. Co-culture techniques have been successfully applied to mimic specific tissues. In 2011, Slater et al. used the co-culturing method to generate a functional tri-layer in vitro model of the glomerular filtration barrier, comprising podocytes and glomerular epithelial cells on an artificial membrane [118].

More complex kidney 3D cell cultures can be also generated from immortalized cells, as is the case in the bioartificial kidney. Janssen et al. have had great success in developing a bioartificial kidney model, by culturing conditionally immortalized PTECs on hollow fiber membranes, resulting in the formation of a tight monolayer, as indicated by the expression of ZO-1. In this model, they could establish the functionality of the OCT2 transporter [119]. Later, the same group generated tubuloids, and adult intestinal stem cell derived epithelial organoids that can be expanded for many passages in growth factor-rich medium, while still remaining genetically stable. The *Lgr5^+^* intestinal stem cells were isolated, and the epithelial elements expanded. This model showed proximal tubule specific gene expression patterns (*ABCC1*, *ABCC3*, *ABCC4*, *SLC22A3*, *SLC40A1*), as well as markers for the collecting duct (*CDH1*, *GATA3*, *AQP3*), loop of Henle (*CLDN10*) and distal tubule (*PCBD1*, *SLC41A3*). The functionality of these tubuloids was established by confirming the functionality of the P-glycoprotein. These tubuloid models consist entirely of epithelial tissue and were used for modeling infectious kidney diseases, kidney malignancies and cystic fibrosis. In these tubuloid models, a stem cell/progenitor state can also be induced to form more heterogenous 3D structures. During this state, the tubuloids express higher levels of CD24, CD44, CD133, SOX9 and Vimentin [120,121]. Recently, tubuloids have been integrated into the organ-on-a-chip system to more closely mimic the tubular microenvironment with extracellular matrix and medium perfusion, a technique that excludes the need for an artificial membrane for cell attachment [122].

## 5. Utility of Urine-Derived Epithelial Cells as Models for Genetic Kidney Diseases

Epithelial cellular models have been valuable tools in the arsenal to study pathophysiological aspects of genetic renal diseases for so long. In most studies, they are usually the essential first step before experimenting with animal models. The type of representative cells in culture, whether human or animal and the approaches to obtain them vary among studies. For podocytes and PTECs, kidney-derived cells from a biopsy were the most common route of obtaining cells for primary culture. This was usually followed by immortalization and knocking-out the target gene to create the disease model of interest. However, when patients’ urine-derived cells were introduced, they gained a lot of interest for their apparent advantages. Urine-derived epithelial cells are easily obtained and propagated in culture, they faithfully represent the disease and the patients they were derived from better than knocked-out models and they are more suitable for experimenting with new therapeutic modalities particularly when immortalized. Table 1 and Table 2 summarize the various types of published cellular models representing genetic diseases associated with podocyte and proximal tubular cellular pathology, respectively.

Revealing novel pathophysiological aspects of genetic renal syndromes is an important field of research, in which epithelial cell models were extensively used in recent years, particularly the urine-derived cells. Rossi et al. investigated the expression of the Nucleotide-binding oligomerization domain, Leucine rich Repeat and Pyrin domain containing (NLRP) family members in human conditionally immortalized PTECs derived from the urine of cystinosis patients. They discovered the essential role of the inflammasome protein NLRP2 in regulating proinflammatory, profibrotic and antiapoptotic responses in cystinotic PTECs, through NF-κB activation [193]. Ivanova et al. studied ATP-induced, IP3-induced and lysosomal calcium release in human PTECs derived from the urine of controls and cystinotic patients and reported the sensitization of ATP-induced calcium release, but with no major dysregulation of intracellular calcium dynamics in cystinotic cells [210]. Uzureau et al. studied conditionally immortalized podocytes derived from the urine of patients with G1 and G2 *APOL1* genotypes. They reported that APOL1 C-terminal variants may induce kidney disease by preventing APOL3 from activating PI4KB, with consecutive actomyosin reorganization of podocytes [129].

Proper confirmation of the genetic etiology in suspected but undiagnosed patients is yet another recent use of urine-derived epithelial cells. Pinto et al. reported the significance of urine-derived podocytes as an important source for genomic DNA for the detection of cryptic mosaicism in X-linked Alport syndrome in a mother that was supposed to donate her kidney to her child. Although her peripheral blood DNA was devoid of pathogenic variants, the DNA derived from podocytes revealed a frame-shift pathogenic variant in *COL4A5* gene confirming the Alport syndrome diagnosis in the mother. This should prompt the reevaluation of living-donor kidney transplantation guidelines in various genetic kidney diseases [211]. Furthermore, the pathogenicity of splicing variants could be easily confirmed at the transcript level in primary podocytes derived from urine of Alport syndrome patients with intronic variants in *COL4A3*, *COL4A4* or *COL4A5* genes [139].

Another major aspect of in vitro studies concerning the kidney is the ability of cellular models to predict the potential human kidney response to various drugs. These studies include the evaluation of potential drug-induced toxicity over the kidney, examination of drug–drug interactions and/or the evaluation of therapeutic effects of the drug in a certain condition. Renal proximal tubules are major players orchestrating nutrient retrieval and waste disposal from the body, including most drugs and their metabolites. Thus, acute or chronic nephrotoxicity is a huge concern during the development process of any drug; particularly that renal insult is often recognized late during clinical trials in many cases [110]. Genetic kidney diseases are particularly sensitive to the study of novel therapeutic agents, as the kidneys are already impacted by their primary disease. Hence, faithful in vitro and in vivo models of the genetic disorder are indispensable when a new drug is introduced.

Moreover, another advantage of cellular models of genetic diseases is their versatility and ability to serve as a target for high-throughput drug screenings. De Leo et al. screened the Prestwick chemical library including 1200 small molecular weight compounds in a urine-derived ciPTEC model of nephropathic cystinosis looking for drugs that reduce levels of the autophagy-related protein p62/SQSTM, since altered autophagy is a major contributing factor to the pathophysiology of the disease. Of the 46 compounds significantly reducing p62/SQSTM1 levels in cystinotic cells, they chose luteolin to study further based on its efficacy and safety profiles. In their study, luteolin had anti-apoptotic properties, was a powerful antioxidant and could stimulate endocytosis through enhancing the expression of the endocytic receptor megalin, which are all beneficial effects for the cystinotic kidney [107].

## 6. Conclusions and Future Perspectives

In conclusion, the urine-derived podocytes and PTECs offer many advantages over established cellular models derived from the kidney. They are normally shed in urine, thus no need to extract by a biopsy procedure. Developing de novo genetic disease models is simply done by extracting the cells directly from affected patients and propagating them in culture after characterization. No need to use an in vitro knocking-out procedure, with all the efficiency issues to worry about during establishing the knock-out model. Urine-derived podocytes and PTECs are fully viable and functional and can represent the target disease faithfully if the proper clone is selected during the characterization process.

Although the numbers of urine-derived podocyte and PTEC models developed to study genetic kidney diseases are rapidly increasing over the last few years, most diseases affecting these cells still lack representative urine-derived models [6,8,100,107]. With the new era of personalized medicine in genetic kidney diseases, not only uncovering the genetic background and linking this genotype to its clinical phenotype is required, but also investigating the individualized response of each patient to various management strategies, in what is known as personalized therapy [17,212,213]. Relevant cellular models can be excellent targets for such research, particularly those obtained directly from diseased individuals in a non-invasive way, such as urine.

Recently, trials for gene therapy and cellular therapy of genetic kidney diseases have been conceptualized in animal models [214,215,216]. These trials are particularly useful in podocyte loss syndromes, since glomerular podocytes are terminally differentiated epithelial cells and unable to proliferate. Human urine-derived renal progenitor cells are a good potential source of podocyte progenitors for cell therapy applications, as they can differentiate in vivo at sites of injured glomeruli into de novo podocytes after intravenous administration [10,11,217].

## Figures and Tables

**Figure 1 cells-10-01413-f001:**
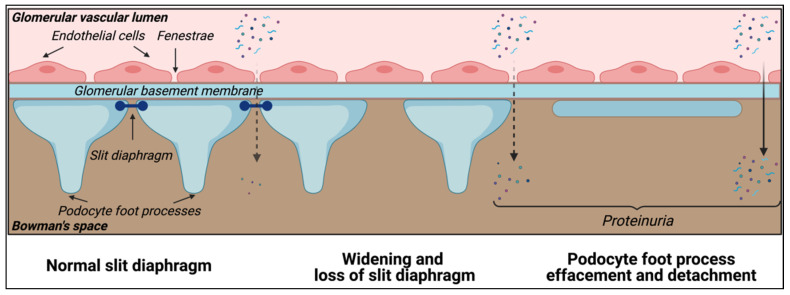
Podocytes: key element of the glomerular filtration barrier. Podocytes are highly differentiated cells with a specific cellular architecture, comprising primary and secondary foot processes that wrap around the glomerular capillaries, and express essential proteins that define the slit diaphragm and permselectivity of the glomerular filtration barrier. Kidney diseases affecting the podocytes (podocytopathies) can be characterized by the loss of the slit diaphragm, podocyte foot process effacement and detachment, which clinically manifests as glomerular proteinuria.

**Figure 2 cells-10-01413-f002:**
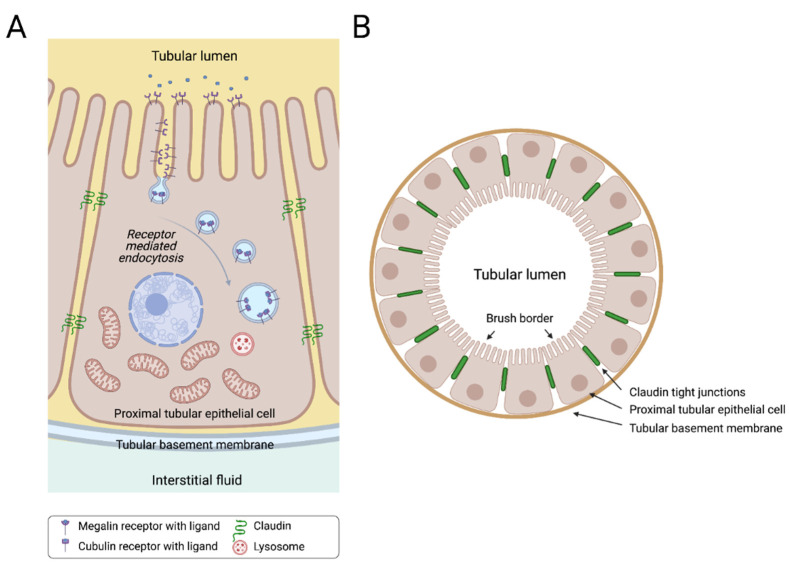
Sagittal (**A**) and cross (**B**) sections of a human proximal tubule. Proximal tubular epithelial cells (PTECs) are indispensable cells for the nephron tubular system. PTECs are highly metabolically active cells that are responsible for reabsorbing many of the filtered solutes in the proximal part of the nephron. These cells are typically characterized by a brush border at the apical side, and high numbers of mitochondria at the basal side of the cell. Megalin and cubulin are characteristic multi-ligand receptors for PTECS, located at the brush border, responsible for receptor-mediated endocytosis for various ligands.

**Figure 3 cells-10-01413-f003:**
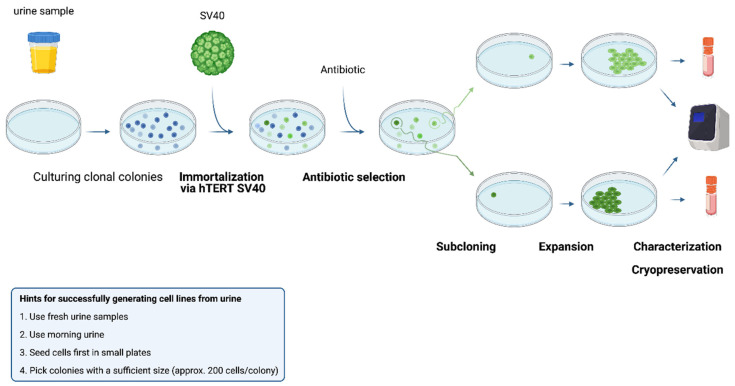
Procedure of isolation, immortalization and subcloning of urine-derived kidney epithelial cells. Fresh urine is cultured in a specific medium that selects for specific types of kidney epithelial cells at 37 °C. Immortalization is performed by use of SV40 and the hTERT gene, upon which antibiotic selection is used to isolate the successfully immortalized cells. Clonal colonies are selected, in order to isolate and expand specific cell clones that will be characterized by biomarker expression and stored for cryopreservation.

**Figure 4 cells-10-01413-f004:**
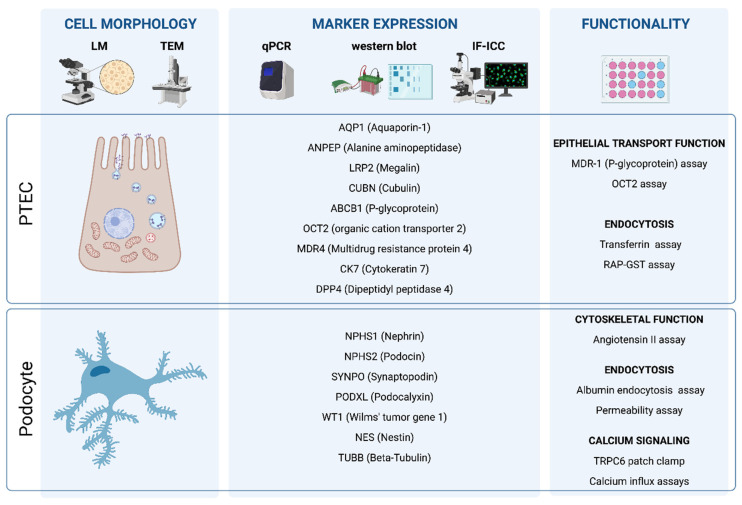
Technology toolbox for characterization of urine-derived kidney epithelial cells. Urine-derived kidney epithelial cells are characterized based on their cellular morphology, the expression of cell-type specific markers, and their functionality via cell-specific assays. To this end, microscopy, including light microscopy (LM) and transmission electron microscopy (TEM), qPCR and Western Blot plus immunocytochemistry are useful technologies.

**Table 1 cells-10-01413-t001:** Cell models of genetic podocytopathies.

Gene (OMIM)	Disease (OMIM)	Inheritance	Origin of Podocyte Models
*ACTN4* (604638)	Focal segmental glomerulosclerosis 1 (603278)	AD	- Human kidney [123] - Mouse kidney [123,124,125]
*ANLN* (616027)	Focal segmental glomerulosclerosis 8 (616032)	AD	- Human kidney [126,127]
*APOL1* (603743)	Focal segmental glomerulosclerosis 4, susceptibility to (612551)		- Human urine [128] - Human kidney [129,130]
*ARHGAP24* (610586)	Focal segmental glomerulosclerosis	AD	- Human kidney [131]
*ARHGDIA* (601925)	Steroid resistant nephrotic syndrome, type 8 (615244)	AR	- Human kidney [132] - Mouse kidney [133]
*AVIL* (613397)	Steroid resistant nephrotic syndrome, type 21 (618594)	AR	- Human kidney [134]
*CD2AP* (604241)	Focal segmental glomerulosclerosis, 3 (607832)	AD	- Human kidney [81,135] - Mouse kidney [81,135,136]
*CDC42* (116952)	Takenouchi–Kosaki syndrome (616737)	AD	- Mouse kidney [135]
*CDK20* (610076)	Steroid resistant nephrotic syndrome	AR	- Human kidney [136]
*CLCN5* (300008)	Dent disease (300009)	XLR	- Human kidney [137]
*COL4A3* (120070)	Alport syndrome 2 (203780) Alport syndrome 3 (104200)	AR AD	- Human urine [8,138]
*COL4A4* (120131)	Alport syndrome 2 (203780)	AR	- Human urine [139,140]
*COL4A5* (303630)	Alport syndrome 1, X-linked (301050)	XLD	- Human urine [8] - Human kidney [141]
*COQ6* (614647)	Coenzyme Q10 deficiency, primary, 6 (614650)	AR	- Human kidney [141,142]
*COQ8B* (615567)	Steroid resistant nephrotic syndrome, type 9 (615573)	AR	- Human kidney [143,144]
*CRB2* (609720)	Focal segmental glomerulosclerosis 9 (616220)	AR	- Human kidney [145]
*CTNS* (606272)	Nephropathic cystinosis (219800)	AR	- Human urine [6] - Human kidney [6]
*CUBN* (602997)	Proteinuria, chronic benign (618884)	AR	- Human kidney [146]
*DAAM2* (606627)	Steroid resistant nephrotic syndrome	AR	- Human kidney [147]
*DGKE* (601440)	Steroid resistant nephrotic syndrome, 7 (615008)	AR	- Human kidney [148]
*DLC1* (604258)	Steroid resistant nephrotic syndrome	AR	- Human kidney [149]
*EMP2* (602334)	Steroid resistant nephrotic syndrome, 10 (615861)	AR	- Human kidney [150,151]
*FAT1* (600976)	Glomerulotubular nephropathy	AR	- Human kidney [152]
*GLA* (300644)	Fabry disease (301500)	XL	- Human urine [153] - Human kidney [154]
*INF2* (610982)	Focal segmental glomerulosclerosis, 5 (613237)	AD	- Human kidney [155] - Mouse kidney [156,157]
*ITGA3* (605025)	Interstitial lung disease, nephrotic syndrome, and epidermolysis bullosa, congenital (614748)	AR	- Mouse kidney [158]
*ITSN1* (602442)	Steroid resistant nephrotic syndrome	AR	- Human kidney [149]
*ITSN2* (604464)	Steroid resistant nephrotic syndrome	AR	- Human kidney [149]
*KANK1* (607704)	Cerebral palsy, spastic quadriplegic, 2 (612900)	AD/AR	- Human kidney [159]
*KANK2* (614610)	Steroid resistant nephrotic syndrome, 16 (617783)	AR	- Human kidney [159]
*KANK4* (614612)	Steroid resistant nephrotic syndrome	AR	- Human kidney [159]
*LAGE3* (300060)	Galloway–Mowat syndrome 2 (301006)	XLR	- Human kidney [160]
*LAMB2* (150325)	Steroid resistant nephrotic syndrome, type 5, with or without ocular abnormalities (614199)	AR	- Mouse kidney [161]
*LMX1B* (602575)	Focal segmental glomerulosclerosis 10 (256020) Nail-patella syndrome (161200)	AD AD	- Human kidney [162] - Mouse kidney [163]
*MAGI2* (606382)	Steroid resistant nephrotic syndrome, type 15 (617609)	AR	- Human kidney [149]
*MYH9* (160775)	Macrothrombocytopenia and granulocyte inclusions with nephritis (155100)	AD	- Human kidney [164] - Mouse kidney [165]
*MYO1E* (601479)	Focal segmental glomerulosclerosis, 6 (614131)	AR	- Human kidney [166,167] - Mouse kidney [168]
*MYO9A* (604875)	Focal segmental glomerulosclerosis	AD	- Mouse kidney [169]
*NPHS1* (602716)	Steroid resistant nephrotic syndrome, type 1 (256300)	AR	- Human kidney [170]
*NPHS2* (604766)	Steroid resistant nephrotic syndrome, type 2 (600995)	AR	- Human urine [171] - Human kidney [171,172]
*NUP85* (170285)	Steroid resistant nephrotic syndrome, type 17 (618176)	AR	- Human kidney [173]
*NUP93* (614351)	Steroid resistant nephrotic syndrome, type 12 (616892)	AR	- Human kidney [174]
*NUP107* (607617)	Steroid resistant nephrotic syndrome, type 11 (616730) Galloway–Mowat syndrome 7 (618348)	AR AR	- Human kidney [173]
*NUP133* (607613)	Steroid resistant nephrotic syndrome, type 18 (618177) Galloway–Mowat syndrome 8 (618349)	AR AR	- Human kidney [173]
*NUP160* (607614)	Steroid resistant nephrotic syndrome, type 19 (618178)	AR	- Human kidney [173]
*NUP205* (614352)	Steroid resistant nephrotic syndrome, type 13 (616893)	AR	- Human kidney [174]
*OSGEP* (610107)	Galloway–Mowat syndrome 3 (617729)	AR	- Human kidney [160]
*PAX2* (167409)	Focal segmental glomerulosclerosis, 7 (616002)	AD	- Human kidney [175]
*PDSS2* (610564)	Coenzyme Q10 deficiency, primary, 3 (614652)	AR	- Mouse kidney [176]
*PLCE1* (608414)	Steroid resistant nephrotic syndrome, type 3 (610725)	AR	- Human kidney [177]
*PODXL* (602632)	Focal segmental glomerulosclerosis	AD/AR	- Mouse kidney [178]
*SGPL1* (603729)	Steroid resistant nephrotic syndrome, type 14 (617575)	AR	- Human kidney [179]
*SYNPO*	Steroid resistant nephrotic syndrome	AR	- Mouse kidney [180,181]
*TBC1D8B* (301027)	Steroid resistant nephrotic syndrome, type 20 (301028)	XL	- Human kidney [182,183]
*TNS2* (607717)	Steroid resistant nephrotic syndrome	AR	- Human kidney [149]
*TP53RK* (608679)	Galloway–Mowat syndrome 4 (617730)	AR	- Human kidney [160]
*TPRKB* (608680)	Galloway–Mowat syndrome 5 (617731)	AR	- Human kidney [160]
*TRPC6* (603652)	Glomerulosclerosis, focal segmental, 2 (603965)	AD	- Mouse kidney [184,185]
*TTC21B* (612014)	Nephronophthisis 12 (613820)	AD/AR	- Human kidney [186]
*WDR73* (616144)	Galloway–Mowat syndrome 1 (251300)	AR	- Human kidney [187,188]
*WT1* (607102)	Steroid resistant nephrotic, type 4 (256370)	AD	- Human kidney [189]
*XPO5* (607845)	Steroid resistant nephrotic syndrome	AR	- Human kidney [174]

AD, autosomal dominant; AR, autosomal recessive; XL, X-linked.

**Table 2 cells-10-01413-t002:** Cell models of genetic proximal tubular diseases.

Gene (OMIM)	Disease (OMIM)	Inheritance	Origin of PTEC Models
*CLCN5* (300008)	Dent disease (300009)	XLR	- Human urine [100] - Human kidney [190] - Mouse kidney [191]
*CTNS* (606272)	Nephropathic cystinosis (219800)	AR	- Human urine [192,193,194] - Human kidney [102,195]
*CUBN* (602997)	Imerslund–Grasbeck syndrome 1(261100)	AR	- Human kidney [196]
*EHHADH* (607037)	Fanconi renotubular syndrome 3 (615605)	AD	- Pig kidney [196]
*GATM* (602360)	Fanconi renotubular syndrome 1 (134600)	AD	- Pig kidney [197]
*HNF4A* (600281)	Fanconi renotubular syndrome 4, with maturity-onset diabetes of the young (616026)	AD	- Human kidney [198]
*LRP2* (600073)	Donnai–Barrow syndrome (222448)	AR	- Human kidney [199]
*OCRL* (300535)	Lowe syndrome (309000) Dent disease type 2 (300555)	XLR XLR	- Human kidney [200,201] - Mouse kidney [191]
*SLC3A1* (104614)	Cystinuria (220100)	AD/AR	- Human kidney [202]
*SLC4A4* (603345)	Renal tubular acidosis, proximal, with ocular abnormalities (604278)	AR	- Human kidney [203]
*SLC4A5* (606757)	Renal tubular acidosis, proximal	AR	- Human urine [204]
*SLC6A19* (608893)	Iminoglycinuria (242600) Hyperglycinuria (138500)	AD/AR AD	- Human kidney [205]
*SLC7A7* (603593)	Lysinuric protein intolerance (222700)	AR	- Human kidney [206]
*SLC7A9* (604144)	Cystinuria (220100)	AD/AR	- Human kidney [205]
*SLC34A1* (182309)	Fanconi renotubular syndrome 2 (613388)	AR	- Human kidney [207]
*SLC36A2* (608331)	Iminoglycinuria (242600) Hyperglycinuria (138500)	AD/AR AD	- Canine kidney [208]
*TRPC3* (602345)	Spinocerebellar ataxia 41 (616410)	AD	- Pig kidney [209]

AD, autosomal dominant; AR, autosomal recessive; XL, X-linked.

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
