# Peer review of "Urine-Derived Epithelial Cells as Models for Genetic Kidney Diseases"

_cells, 2021, doi:10.3390/cells10061413_

Round 1

Reviewer 1 Report

This review discussed the use of urine-derived epithelial cells in the study of genetic kidney diseases. It includes a substantial review of every aspects regarding quantification urine derived cells, isolation and immortalization techniques, characterization of the cells, and the utility of the urine-derived cells as models for genetic kidney diseases. Overall, this is an outstanding review. However, the quality of the images presented in the paper are not optimal. Higher resolution should be applied in these images. Especially the figure 4, which is very blurry and the texts are difficult to be recognized. 

Author Response

Comments and Suggestions for Authors

This review discussed the use of urine-derived epithelial cells in the study of genetic kidney diseases. It includes a substantial review of every aspects regarding quantification urine derived cells, isolation and immortalization techniques, characterization of the cells, and the utility of the urine-derived cells as models for genetic kidney diseases. Overall, this is an outstanding review. However, the quality of the images presented in the paper is not optimal. Higher resolution should be applied in these images. Especially the figure 4, which is very blurry and the texts are difficult to be recognized.

  • We thank the reviewer. The conversion of the previously submitted file to pdf format probably decreased the quality of the images. Based on the reviewer's comment, in combination with the manuscript we have provided high quality images for our figures as separate submissions to be used by the journal.
  • As for Figure 4, we have enhanced the text size and quality. The dimensions of the figure also need to be increased in its corresponding page to become much clearer. 

Reviewer 2 Report

The present propose a well-structured and synthesized review,  which is rich in information with reference elements concerning the methodology for establishing human podocyte and PTEC cell lines and highlight their importance as kidney disease cell models. The authors present the well-established and recent cell isolation techniques, quantification, immortalization and characterization, and their applications.

The whole work is coherent.

Author Response

Comments and Suggestions for Authors

The present propose a well-structured and synthesized review, which is rich in information with reference elements concerning the methodology for establishing human podocyte and PTEC cell lines and highlight their importance as kidney disease cell models. The authors present the well-established and recent cell isolation techniques, quantification, immortalization and characterization, and their applications. The whole work is coherent.

- We thank the reviewer.

Reviewer 3 Report

In this submitted review work by Bondue et al. are described in details the newest advances on urine-derived epithelial cells of kidney origin, their isolation, characterization and their potential for in vitro disease modeling. 
Overall, the work is well written and easy to read also for a broad audience. Listed below are the few issues that should be tackled before moving forward with publication. 

1) In section 4, subcloning is mentioned but this part should be expanded a little to describe methodology and criteria for cell selection. 

2) in 4.1 and 4.2, characterization of podocytes and tubular cells is reported, and positive results have been reported. However there is no mention of important characteristics for both cell lines like expression of collagen IV alpha chains (for podocytes) or OAT transporters (for proximal tubular cells). Whether those analysis were not performed or resulted negative, they should be reported and critically discussed to provide the readers with all the information

3) in the first paragraph of 4.3, the Authors discuss viability and functionality of urine-derived cells and to support their claim, they report the fact that even in healthy individuals, cells are found in the urine. However, it should be make clear that it is not known if cells that are shedding into the urine from healthy individuals have suffered some damaged leading to their detachment or not. We cannot beforehand assume that they are healthy and this should be stated.

4) The last paragraph of section 4.4 (Lines 537-555) does not fit into the overall theme of the current manuscript. It is not clear how description of hiPSC-based organoids fits into the current manuscript. My suggestion would be to remove it for clarity. 

Author Response

Comments and Suggestions for Authors

In this submitted review work by Bondue et al. are described in details the newest advances on urine-derived epithelial cells of kidney origin, their isolation, characterization and their potential for in vitro disease modeling. 
Overall, the work is well written and easy to read also for a broad audience. Listed below are the few issues that should be tackled before moving forward with publication. 

1) In section 4, subcloning is mentioned but this part should be expanded a little to describe methodology and criteria for cell selection.

  • Based on the reviewer's comment, we have added a paragraph discussing the subcloning process in more details (Page 8, line 329 of the revised tracked manuscript): "Once cells have been immortalized, a homogenous cell line can be obtained by subcloning[78]. Subcloning is usually done by using irradiated NIH 3T3 mouse fibroblast cells as non-dividing feeder cells. Briefly, cells are seeded at densities of 100, 200, 300 and 400 cells per 25cm² flask and grown at 33°C. Subsequently, feeder cells are added to each flask at a density of 0.5 x 106 cells/flask. After being cultured for about 21-28 days, clones derived from single cells become visible and are picked by using cloning discs drained in trypsin/EDTA and transferred to individual wells of a 24 well plate for expansion. At this stage, usually the fast growing colonies are selected to be expanded further. Finally, when the cells are grown to confluence, they are further transferred to larger flasks, and each clone can be either cryopreserved or processed for characterization[70,77]. It is worth mentioning that subcloning of urinary kidney epithelial cells can also be done without feeder cells. Briefly, cells are seeded into a 96 well plate at densities of 0.5 and 1 cell per well. After 14 days, colonies derived from single cells become visible, and can be expanded[79]."

  • There are no specific criteria for the selection of cells during subcloning, just the fast growing colonies are usually selected at this stage. As for the criteria of cellular selection after subcloning, it is based on the molecular and functional characterization of the resulting subclones, which is discussed extensively in sections 4.1. and 4.2. for podocytes and PTECs, respectively.

2) in 4.1 and 4.2, characterization of podocytes and tubular cells is reported, and positive results have been reported. However there is no mention of important characteristics for both cell lines like expression of collagen IV alpha chains (for podocytes) or OAT transporters (for proximal tubular cells). Whether those analysis were not performed or resulted negative, they should be reported and critically discussed to provide the readers with all the information.

  • We thank the reviewer. We added few sentences discussing the expression of collagen IV alpha chains in podocytes (Page 10, line 390 of the revised tracked manuscript): "The attachment of podocytes to the glomerular basement membrane is granted through various proteins and the production and assembly of collagen type IV α-chains (COL4A3, COL4A4 and COL4A5). The expression of collagen IV α3, α4 and α5 is podocyte specific and has been applied for the characterization of mature and functional podocyte cell lines [8,88]. Additionally, in kidney organoids and 3D podocyte cultures, type IV collagen α-chains show higher abundance and are indicative of basement membrane formation[89–93].", and expression of OAT transporters in PTECs (Page 11, line 478 of the revised tracked manuscript): "When using PTEC models for drug testing, the expression of organic anion transporters (OATs) is often desired, including OAT1/SLC22A6 and OAT3/SLC22A8. However, OAT expression is often lacking in 2D cultures of immortalized PTECs, as is the case with HK-2 cell lines. Therefore, some OAT-overexpressing lines have been developed for the investigation of drugs, but the expression can also be ensured by using 3D cell cultures or primary human tubular cell monolayers, both of which have established OAT-expression[109–112]."

3) In the first paragraph of 4.3, the Authors discuss viability and functionality of urine-derived cells and to support their claim, they report the fact that even in healthy individuals, cells are found in the urine. However, it should be make clear that it is not known if cells that are shedding into the urine from healthy individuals have suffered some damaged leading to their detachment or not. We cannot beforehand assume that they are healthy and this should be stated.

  • Based on the reviewer's comment, we added the following sentence to the above mentioned paragraph (Page 12, line 498 of the revised tracked manuscript): "Of note, it is important to consider that healthy donors are those that have not been diagnosed with kidney disease, which does not exclude the possibility of unknown damage leading to cells detachment, thus the processes of cell characterization and selection are extremely important to ensure the functionality of the urine-derived control cells before being used in experiments. In this regard, several studies have compared properly selected control cells from urine with cells isolated directly from the kidney and proved their similarities and functionality, underlining the potential of these cell lines as wild type controls for the study of models of kidney diseases[10,77]"

4) The last paragraph of section 4.4 (Lines 537-555) does not fit into the overall theme of the current manuscript. It is not clear how description of hiPSC-based organoids fits into the current manuscript. My suggestion would be to remove it for clarity.

  • We thank the reviewer. We removed the above mentioned paragraph from the revised version of the manuscript.